# Associations of the Short Physical Performance Battery (SPPB) with Adverse Health Outcomes in Older Adults: A 14-Year Follow-Up from the English Longitudinal Study of Ageing (ELSA)

**DOI:** 10.3390/ijerph192316319

**Published:** 2022-12-06

**Authors:** Max J. Western, Olivia S. Malkowski

**Affiliations:** Centre for Motivation and Health Behaviour Change, Department for Health, University of Bath, Claverton Down, Bath BA2 7AY, UK

**Keywords:** ageing, physical function, physical performance, mobility, disability, falls, health, SPPB, older adults, ELSA

## Abstract

The Short Physical Performance Battery (SPPB) is an objective tool for evaluating three domains (balance, repeated chair stands, and gait speed) of lower extremity physical function in older age. It is unclear how the associations between SPPB scores and health outcomes persist over time. The aim of this 14-year cohort study was to investigate associations between SPPB scores and health outcomes among participants aged 60+ years in the English Longitudinal Study of Ageing (ELSA). The exposures were SPPB scores (total and domain-specific) at baseline (Wave 2). The outcomes were mobility impairments, difficulties in performing basic activities of daily living (ADL) or instrumental activities of daily living (IADL), and falls, measured at seven subsequent timepoints (Waves 3 to 9). The analyses involved linear and logistic multilevel regressions. After adjusting for potential confounders, a one-point increase in the total SPPB score was associated with a 0.13 (95% CI: −0.16, −0.10) decrease in mobility impairment, a 0.06 (−0.08, −0.05) decrease in ADL disabilities, a 0.06 (−0.07, −0.04) decrease in IADL disabilities, and 8% (0.90, 0.95) lower odds of falling (averaged across all follow-ups). Associations between the SPPB domains and health outcomes were more varied. The SPPB may be a useful measure for identifying older adults at a high risk of adverse outcomes.

## 1. Introduction

Biological aging in humans is often characterised by a decline in physical function including balance, muscular strength, and walking speed [1]. Good physical function has been suggested to prevent frailty and underpin people’s ability to remain physically active and carry out the activities of daily living (ADL) that help maintain independence, as well as prevent falls and hospitalisation [2]. Given the rapid aging of the population and the associated strain on health and social care, much attention in contemporary public health policy and research has focused on the relationship between physical function and health to guide preventive interventions that help keep older adults independent for as long as possible [3,4]. Central to the understanding of this relationship is a measure of physical function that can guide researchers and practitioners on the most appropriate action to prevent adverse health outcomes and declining quality of life [5].

The Short Physical Performance Battery (SPPB) [6] has emerged as one of the leading tools for assessing physical function. The SPPB measures three components of physical function: balance, as measured across three levels of difficulty; lower limb strength, measured as the speed at which an individual can perform five unassisted sit-to-stand movements; and gait speed, measured as a 2.44-metre walk. The original study presenting the validity of the SPPB has been cited over 8600 times as of October 2022, and the instrument is often used as a measure to assess an intervention’s effects in trials targeting improved physical function [7,8]. Moreover, the SPPB has been proposed to be an appropriate measure for characterising the frailty status of older adults, distinguishing robust physical function from pre-frailty or frailty [9,10].

In addition to its widespread use as an outcome measure of physical function, previous work has shown that lower scores on the SPPB are associated with a range of adverse outcomes, including falls [11,12], hospitalisation [13,14], long-term care needs [13,15], frailty [16], and all-cause mortality [17]. However, studies with longer follow-up periods providing evidence on how the association between SPPB scores and physical outcomes changes over time are scarce. Likewise, little attention has been spent on ascertaining the relative contribution of the subcomponents of the SPPB for protecting against mobility impairments, loss of the ADL, and falls.

The aim of the present study was therefore to examine associations of the SPPB and its domains of balance, strength, and gait speed with the key functional outcomes collected over 14 years. We hypothesised that higher total SPPB scores at baseline would be associated with more favourable outcomes over time, as would higher values across each of the domains.

## 2. Materials and Methods

### 2.1. Study Design and Participants

We used data from Waves 2 (2004–2005) to 9 (2018–2019) of the English Longitudinal Study of Ageing (ELSA) [18], a nationally representative biannual survey of English adults aged 50+ years, living in private households. The original sample of respondents was recruited from households participating in the Health Survey for England in 1998, 1999, or 2001. Further information on the cohort profile is available elsewhere [19]. Wave 2 was used as the baseline assessment, as the SPPB was first administered in this wave. We limited the sample to core members aged 60+ years at baseline, to align with the World Health Organization’s definition of older age [20]. ELSA received ethical approval from the National Research Ethics Service and all participants provided informed consent. The present study was approved by the Research Ethics Approval Committee for Health (EP 22 048) at the University of Bath. This study has been reported in line with the Strengthening the Reporting of Observational Studies in Epidemiology (STROBE) guidelines (see the checklist in Appendix A) [21].

### 2.2. Measures

#### 2.2.1. Outcomes

There were four outcome variables assessed at every timepoint: (1) mobility impairments, (2) ADL disabilities, (3) instrumental activities of daily living (IADL) disabilities, and (4) falls. Mobility impairments were operationalised as the sum of reported difficulty with 10 activities (0, no difficulty; 10, difficulty performing all activities), namely walking 100 yards; sitting for about 2 h; getting up from a chair after sitting for long periods; climbing several flights of stairs without resting; climbing one flight of stairs without resting; stooping, kneeling, or crouching; reaching or extending arms above shoulder level; pulling or pushing large objects; lifting or carrying over 10 pounds; and picking up a 5 pence coin from a table. Functional decline was evaluated on the basis of self-reported limitations in basic ADL and IADL. There were six ADL items: dressing, walking across a room, bathing or showering, eating, getting in and out of bed, and using the toilet. The seven IADL items included using a map, preparing a hot meal, shopping for groceries, making telephone calls, taking medications, doing work around the house and garden, and managing money. The responses were summed to construct a scale ranging from 0 to 6 for ADL disabilities, and from 0 to 7 for IADL disabilities, representing the number of activities with reported difficulty. Finally, the participants were asked whether they had fallen in the previous 2 years (no versus yes).

#### 2.2.2. Exposures

The SPPB is composed of three measures of physical performance: standing balance, repeated chair stands, and gait speed [6]. Balance was evaluated using three hierarchical tasks involving side-by-side, semi-tandem, and full-tandem stands. For the repeated chair stand test, participants were timed as they completed five sit-to-stand repetitions. Gait speed was assessed by timing the participants as they walked 2.44 metres at their regular pace. Each SPPB domain was scored from 0 (worst) to 4 (best) according to established cut-off points and summed to generate a total score from 0 to 12 [6]. A detailed explanation of the scoring methods is provided in the Appendix A. We used the baseline SPPB scores (total and domain-specific) as the exposures.

#### 2.2.3. Covariates

Potential confounders were selected in line with existing studies [13,22]. All measures were retrieved at baseline. Sociodemographic covariates included age (a continuous variable, collapsed to 90 for participants aged 90+ years), biological sex, ethnicity (dichotomised in ELSA as White versus non-White), marital status (single, separated/divorced/widowed, and married), employment status (not employed versus working full- or part-time), education (no formal qualifications, school qualifications, at least some higher education), and total non-pension wealth (quintiles) at the benefit unit level.

Health-related covariates comprised physical activity, body mass index (BMI), cognitive function, and depressive symptoms. The respondents reported how frequently they participated in sports or activities that were vigorous, moderately energetic, and mildly energetic (more than once a week, once a week, 1 to 3 times a month, hardly ever or never). A four-category variable was created for physical activity, in line with previous work [23]: inactive (no weekly activity), only mild activity at least once per week, moderate but no vigorous activity at least once per week, and vigorous activity at least once per week. BMI was a continuous variable, calculated as weight (kg) divided by height squared (m^2^). Cognitive function was assessed by aggregating information from neuropsychiatric batteries testing time orientation, immediate and delayed recall, prospective memory, verbal fluency, and executive function (processing speed and efficiency) [24,25]. As the scoring of each test varied, we computed 𝓏-scores for the individual tests. The sum of the individual domains’ 𝓏-scores was then standardised to obtain a 𝓏-score for global cognitive function [26]. Depressive symptoms were assessed using the 8-item Centre for Epidemiologic Studies Depression Scale (CES-D) [27].

### 2.3. Statistical Analysis

We defined our complete-case samples as participants with data on the SPPB exposures (total score), covariates, and outcomes at baseline and at least one follow-up wave. Descriptive statistics were calculated as the mean (standard deviation (SD)) for continuous or count variables, and the frequency (percentage) for categorical variables. We compared the unweighted baseline characteristics of the participants with complete and missing data, using independent t-tests and Pearson’s chi-squared (χ^2^) tests.

In our primary analyses, we used linear (mobility impairments, ADL disabilities, IADL disabilities) and logistic (falls) multilevel models to examine the associations between total SPPB scores (Wave 2; continuous exposure) and subsequent outcomes (Waves 3 to 9). The models were built in 10 stages (Appendix A) [26]. The analyses were repeated to explore the associations between the SPPB domain scores (entered as categorical exposures in separate models and then simultaneously in mutually adjusted models) and outcomes previously described (see Appendix A for the stages of the models). Binary variables (no difficulties versus difficulties performing at least one activity) for mobility impairment, ADL disabilities, and IADL disabilities were used as secondary outcomes in the multilevel logistic models.

ELSA has a hierarchical structure, with repeated measures nested within persons. Therefore, we included a random intercept at the individual level and a random slope according to time (defined by the wave of follow-up) in the linear multilevel models. Due to convergence issues encountered when running the random intercept and (random) slope models, random intercept (fixed slope) models were performed for all binary outcomes. The baseline cross-sectional sampling weight was applied to all models to improve population representativeness.

As a sensitivity analysis, multiple imputation with chained equations was conducted to replace missing data on covariates and outcomes, using all exposures (total SPPB score, balance score, repeated chair stand test score, and gait score), outcomes, and covariates as predictors, as well as several baseline auxiliary variables including self-reported general health (1: poor, 5: excellent), occupational class (using the three-class National Statistics Socio-Economic Classification), smoking status (never a smoker, former smoker, current smoker), alcohol consumption (less than once a week, one to four times per week, five or more times per week), living status (living alone versus not living alone), and the presence of any limiting long-standing illness (no versus yes). Predictive mean matching was used to impute the data on mobility impairment, ADL disabilities, IADL disabilities, BMI, and depressive symptoms, according to the “just another variable” approach [28]. Following the recommendations for imputing derived variables (e.g., BMI) [29], height and weight were incorporated into the imputation model. Given the discrepancies in the missing data across cognitive tests, the 𝓏-scores for the individual domains were imputed using a linear regression model [30]. The 𝓏-score of global cognitive function was imputed passively [30]. We included interactions between total SPPB score and age, and between total SPPB score and biological sex to ensure congruence with the primary analytical models [28,30]. The imputation model was weighted using the cross-sectional sampling weight from Wave 2. The patterns of missing data are shown in the Appendix A. The data were assumed to be missing at random, and 25 datasets were imputed and combined for analyses using Rubin’s rules. In a further set of sensitivity analyses, we planned to analyse the three count outcomes (i.e., mobility impairment, ADL disabilities, and IADL disabilities) using multilevel mixed-effects negative binomial regressions. However, because of convergence issues, these models are not presented.

Multilevel models were performed using the “mixed” and “melogit” commands in Stata/BE Version 17.0 (College Station, TX, USA: StataCorp LP). The syntax is openly available at https://github.com/OliviaMalkowski/SPPB-study.git. Statistical significance was defined as *p* < 0.05.

## 3. Results

### 3.1. Descriptive Statistics

Of 6183 eligible participants aged 60+ years at baseline, our complete-case samples for models with mobility impairment, ADL or IADL disabilities, and falls as outcomes included 3548, 3547, and 3505 participants, respectively (Table 1, Appendix A). Descriptive statistics summarising the outcome variables at each of the follow-up waves are presented in Appendix A. Compared with the complete-case samples, participants with missing data were older on average, and a higher proportion were female, of non-White ethnic origin, unmarried, and of lower socioeconomic status (Appendix A). Moreover, the excluded samples had lower cognitive function scores, SPPB scores (total and domain-specific), and BMI values; they also reported more depressive symptoms, mobility impairments, ADL disabilities, and IADL disabilities. In addition, a lower proportion of participants in the analytical samples were physically inactive and had experienced a fall.

### 3.2. Associations of the Total SPPB Score with Adverse Health Outcomes

In the univariate models (Model 1), a one-point increase in the total SPPB score was associated with a 0.55 decrease (95% confidence intervals (CI): −0.58, −0.52) in mobility impairment, a 0.16 decrease (95% CI: −0.18, −0.15) in ADL disabilities, a 0.18 decrease (95% CI: −0.19, −0.16) in IADL disabilities, and 15% lower odds (95% CI: 0.83, 0.87) of falling on average across all follow-up waves (Table 2). There was an increase in mobility impairment, ADL disabilities, IADL disabilities, and the odds of falling over time (Model 2; all *p* < 0.001). The associations were non-linear (Model 3) in the models with ADL and IADL disabilities as outcomes (both *p* < 0.001). There was little change in the associations after adjusting for sociodemographic covariates (Model 4). The associations between the total SPPB score and mobility impairment (−0.36 (95% CI: −0.40, −0.33)), ADL disabilities (−0.12 (95% CI: −0.14, −0.11)), IADL disabilities (−0.11 (95% CI: −0.13, −0.09)), and the odds of falling (0.90 (95% CI: 0.88, 0.93)) remained but were attenuated after further adjustment for health-related covariates (Model 5). The results were statistically significant (all *p* < 0.001) even when accounting for the baseline outcomes (Model 6). We found similar patterns of association when treating mobility impairment, ADL disabilities, and IADL disabilities as binary variables (Appendix A) and conducting multiple imputation analyses (Appendix A).

There was a slightly greater decrease (interaction term: −0.004 (95% CI: −0.006, −0.002)) in IADL disabilities per one-point increase in the total SPPB score for older participants in our sample (Model 7). However, there was no evidence that the associations between the total SPPB score, and mobility impairment, ADL disabilities, or falls were modified by age (all *p* > 0.05). For each one-point increase in the total SPPB score, the decrease in mobility impairment (interaction term: 0.06 (95% CI: 0.02, 0.11)) and the odds of falling (interaction term: 1.07 (95% CI: 1.02, 1.12)) was smaller among women than men, on average across all follow-ups (Model 8).

The magnitude of the associations between the total SPPB score and mobility impairment (−0.02 (95% CI: −0.02, −0.01)), ADL disabilities (−0.02 (95% CI: −0.03, −0.01)), and IADL disabilities (−0.04 (95% CI: −0.05, −0.03)), but not falls (*p* = 0.544), increased as follow-up progressed (Model 9). We found evidence of a non-linear influence of time (Model 10) on the association between the total SPPB score and mobility impairment (*p* = 0.013 for the interaction between the total SPPB score and quadratic time). These interactions were statistically significant in the mutually adjusted models (Model 11; Figure 1 and Appendix A).

No statistically significant interactions emerged in the multilevel logistic models with mobility impairment or IADL disabilities as outcomes. However, in addition to a statistically significant interaction with linear time (*p* = 0.005), the decrease in the odds of reporting one or more ADL disabilities for each one-point increase in the total SPPB score was smaller among older respondents (interaction term: 1.007 (95% CI: 1.003, 1.012); Model 11) in the multilevel logistic model. The results of multiple imputation were comparable with the primary analyses. An exception was the interaction between the total SPPB score and time, which was no longer statistically significant (*p* = 0.079) in the model with mobility impairment as the outcome (Model 9). Moreover, there was evidence of an interaction between the total SPPB score and quadratic time (0.005 (95% CI: 0.003, 0.008)) on IADL disabilities (Models 10 and 11).

### 3.3. Associations of the SPPB Domains with Adverse Health Outcomes

In the unadjusted models (Model 1; Table 3), balance scores of two or more were associated with significantly fewer mobility impairments and IADL disabilities (all *p* ≤ 0.01), relative to individuals with zero points (the reference class). There was a clear gradient in ADL disabilities from the highest to the lowest balance score (all *p* ≤ 0.001). Scoring three (odds ratio (OR) = 0.60, *p* = 0.042) or four (OR = 0.33, *p* < 0.001) points on the balance test was associated with significantly lower odds of falling. Gait scores of two or more were associated with fewer mobility impairments and ADL disabilities, and lower odds of falling (all *p* < 0.05); participants scoring three or four points on the gait test reported fewer IADL disabilities (both *p* < 0.001).

In the fully adjusted models (Model 6), balance was not associated with mobility impairment. Only participants scoring four points on the balance test had significantly fewer IADL disabilities (−0.36 (95% CI: −0.65, −0.06)) and reduced odds of falling (0.58 (95% CI: 0.37, 0.89)). Balance scores of two or more were associated with fewer ADL disabilities (all *p* < 0.01). Three or four points on the gait test were associated with fewer mobility impairments (both *p* < 0.05), and four points were associated with fewer (−0.42 (95% CI: −0.68, −0.17)) IADL disabilities. Higher gait scores were associated with significantly fewer ADL disabilities and decreased odds of falling (all *p* < 0.05). Sit-to-stand performance was associated with all outcomes in the expected directions in the unadjusted and fully adjusted models (all *p* < 0.01).

When the SPPB domains controlled for one another, the results were largely maintained (Appendix A). However, in the unadjusted models, only a balance score of four was associated with significantly fewer IADL disabilities (*p* = 0.007). Balance was not associated with mobility impairment or falls. Furthermore, only participants with gait scores of three or four had significantly fewer mobility impairments and lower odds of falling (all *p* ≤ 0.001). In the fully adjusted models, balance and sit-to-stand scores were no longer associated with IADL disabilities or falls. A gait score of four was associated with a 0.70 (95% CI: −1.07, −0.33) decrease in mobility impairments and a 0.33 (95% CI: −0.59, −0.07) decrease in IADL disabilities. Participants scoring one, three, or four points on the gait test reported fewer ADL disabilities, and those scoring three or four points showed reduced odds of falling, relative to the reference groups (all *p* < 0.05). These patterns of the findings were broadly similar when using binary outcomes (Appendix A) and multiple imputation (Appendix A), although the statistical significance levels varied.

## 4. Discussion

In this study, we provide evidence that the SPPB, as a global measure of physical function, is consistently associated with favourable outcomes relating to robust physical health, including the prevention of mobility impairment, ADL disabilities, and IADL disabilities, as well as lower odds of falling. These associations remained across seven waves of follow-up spanning 14 years, even after adjusting for demographic and health-related covariates. The domain-specific results were more varied, whereby only lower extremity muscle strength, as measured by repeated chair stands, was consistently independently associated with favourable outcomes, although the statistical significance levels varied between the unadjusted and fully adjusted models. For balance and gait speed, only scores at the highest end of the respective scales were associated with better outcomes relative to the lowest (most unfavourable) possible score.

The present study, importantly, provides evidence for the discriminant validity of the SPPB, that is, its ability to distinguish between people with high and low function, in line with their differing functional capabilities. The finding that the total SPPB score was associated with favourable outcomes is consistent with other studies that have explored associations between physical function and a range of physical health outcomes [11,12,13,14,16,31], including difficulty performing ADL and IADL [13,14]. The results, however, contrast with those of a recent prospective study in Sweden, which showed that the SPPB score was not associated with falls in 202 older adults aged 75 years or older [32]. These discrepancies might be explained by the smaller sample size, the shorter 1-year follow-up, or the high mean (SD) baseline SPPB score of 10.7 (1.4) and the consequent limited variability in the Swedish study for detecting meaningful differences.

An interesting finding of the present study relates to the incongruent results amongst the subcomponents of the SPPB tool. We hypothesised that each domain of balance, strength, and gait speed would be independently associated with the outcomes of interest. However, this was not entirely the case. Differences in the performance of the subcomponents may be explained by the way they assess the respective aspects of physical function. The measure of strength, which asks the participant to perform five sit-to-stand repetitions as fast as they can, is the only one of the three independent assessments to measure maximal capability. The balance task, by contrast, has three levels of difficulty but the test ceases if a participant holds each position for 10 s, with individuals being assigned maximum points for achieving all three. Indeed, a top score of 4 was by far the most frequent score amongst the analysed cohort, accounting for over 78% of participants. Similarly, the gait speed test, which asks participants to walk at their normal, rather than top, speed also does not require maximal effort. The frequency of achieving a top score for gait speed was 68% in the current sample, compared with 40% for the chair rise test.

These findings suggest that the SPPB, when used as intended as a composite measure of physical function, is a useful tool for identifying individuals who might be more prone to falling or losing independence. Consistent with previous work, the analysis of this large dataset demonstrated that higher overall function scores may have a protective influence on physical health, both in the short and long term [17]. The strength component of the SPPB appears to be useful as an independent assessment of that domain. However, for independent assessments of balance or gait speed, researchers may look to other more sensitive measures to discriminate older adults’ capabilities in these components of physical function. Such examples might be the Community Balance and Mobility Scale, which has been shown to overcome issues with ceiling effects pertaining to the SPPB [33], or a fast, rather than normal, gait speed test [34]. Nonetheless, our results suggest the SPPB can be used as a measure for helping clinical practitioners or researchers identify individuals who would benefit from interventions targeting improved physical function. 

A strength of the present work is the use of data from a large longitudinal cohort study with long-term follow-up that allows an assessment of the associations between physical function and physical health over time. This study also benefitted from a rigorous analysis of the SPPB and its domains using statistical models assembled in several stages. Still, the analysis may be limited somewhat by the homogenous sample in terms of ethnicity (although it was diverse across certain markers of socioeconomic position), self-reported outcomes that may be prone to bias, and survival bias within the study population as reflected by the observed attrition across timepoints, which, given the age of the population and the length of follow-up, was not unexpected. Moreover, while zero-inflated negative binomial models have been recommended for analysing count outcomes (e.g., mobility impairment, ADL disabilities, IADL disabilities) with excess zeros, these models were not considered in the present work due to software constraints precluding the specification of a multilevel framework [35]. Nevertheless, these models are likely to produce less biased estimates than linear or logistic regression models, and may be better suited for the investigation of exposures associated with count outcome measures, such as mobility impairment, ADL disabilities, or IADL disabilities [35].

## 5. Conclusions

Overall, this study demonstrated that the SPPB was associated with adverse future physical health outcomes in a sample of older adults in England. As hypothesised, we observed that higher overall physical function, as measured by the total SPPB score, was associated with the maintenance of mobility, retention of the ability to complete ADL and IADL, and lower odds of falling. While the SPPB-measured chair rise test appears to be a useful measure of lower extremity strength, for isolated assessments of balance or gait speed, researchers and practitioners may be advised to seek more robust and sensitive measures.

## Figures and Tables

**Figure 1 ijerph-19-16319-f001:**
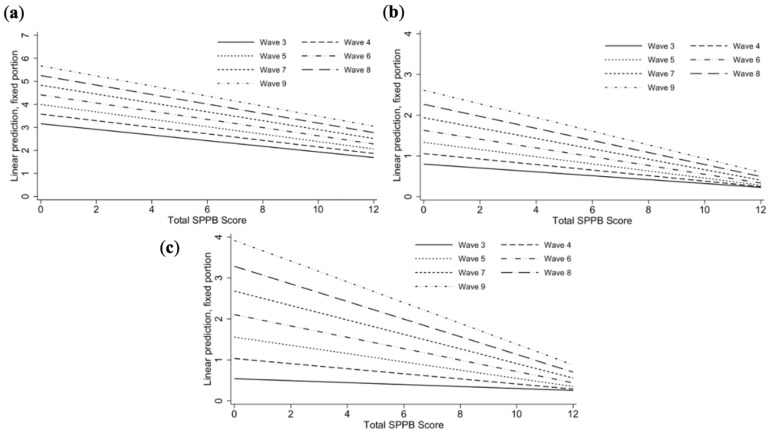
Simple slopes for the relationship between the total SPPB score at baseline and mobility impairment (**a**), ADL disabilities (**b**), and IADL disabilities (**c**) at different follow-up waves. Note: The predicted margins were derived from Model 11 for mobility impairment (**a**), Model 9 for ADL disabilities (**b**), and Model 11 for IADL disabilities (**c**). The values were computed using the postestimation “margins” command with the “vce(unconditional)” option. The random effects of the respective multilevel models were fixed to zero. The simple slopes show the amount of change in each outcome for a one-point increase in the total SPPB score at baseline while holding time (wave of follow-up) constant at different values.

**Table 1 ijerph-19-16319-t001:** Baseline characteristics of the complete-case samples.

	Mobility Impairments (*n* = 3548)	ADL/IADL Disabilities (*n* = 3547)	Falls (*n* = 3505)
Age, mean (SD) ^1^	70.6 (7.6)	70.6 (7.6)	70.5 (7.6)
Biological sex, *n* (%)			
Male	1646 (46.7)	1646 (46.7)	1628 (46.9)
Female	1902 (53.3)	1901 (53.3)	1877 (53.2)
Ethnicity, *n* (%)			
White	3502 (98.4)	3501 (98.4)	3460 (98.4)
Non-White	46 (1.6)	46 (1.6)	45 (1.6)
Marital status, *n* (%)			
Single, i.e., never married	159 (4.7)	159 (4.7)	157 (4.7)
Legally separated	31 (1.0)	31 (1.0)	31 (1.0)
Divorced	256 (6.8)	256 (6.8)	256 (6.9)
Widowed	803 (22.9)	803 (22.9)	792 (22.8)
Married, first and only marriage	1969 (55.6)	1968 (55.6)	1944 (55.6)
Remarried, second or later marriage	330 (9.0)	330 (9.0)	325 (9.0)
Employment, *n* (%)			
Retired	2599 (72.9)	2598 (72.9)	2568 (73.0)
Unemployed	10 (0.3)	10 (0.3)	10 (0.3)
Permanently sick or disabled	91 (2.7)	91 (2.7)	90 (2.7)
Looking after home or family	344 (9.9)	344 (9.9)	337 (9.9)
Semi-retired	23 (0.6)	23 (0.6)	23 (0.6)
Employed	358 (10.0)	358 (10.0)	357 (10.2)
Self-employed	123 (3.4)	123 (3.4)	120 (3.4)
Education, *n* (%)			
No qualification	1589 (48.1)	1588 (48.1)	1564 (47.8)
Secondary or lower	1036 (28.2)	1036 (28.2)	1026 (28.3)
Higher education below degree	485 (12.7)	485 (12.7)	480 (12.7)
Degree or equivalent	438 (11.1)	438 (11.1)	435 (11.2)
Wealth, *n* (%)			
First quintile (lowest)	551 (17.0)	551 (17.0)	544 (17.1)
Second quintile	663 (18.7)	663 (18.7)	660 (18.8)
Third quintile	712 (20.2)	712 (20.2)	702 (20.1)
Fourth quintile	784 (21.7)	784 (21.7)	771 (21.6)
Fifth quintile (highest)	838 (22.4)	837 (22.4)	828 (22.4)
Physical activity, *n* (%)			
Inactive	221 (6.5)	220 (6.5)	211 (6.2)
Mild activity	528 (15.5)	528 (15.5)	521 (15.5)
Moderate activity	1816 (51.3)	1816 (51.3)	1796 (51.4)
Vigorous activity	983 (26.7)	983 (26.7)	977 (26.9)
BMI (kg/m^2^), mean (SD)	27.8 (4.6)	27.8 (4.6)	27.9 (4.6)
Cognitive function, mean (SD)	0.0 (1.0)	0.0 (1.0)	0.0 (1.0)
Depressive symptoms (range: 0–8), mean (SD)	1.4 (1.8)	1.4 (1.8)	1.4 (1.8)
Total SPPB score (range: 0–12), mean (SD)	9.7 (2.7)	9.7 (2.7)	9.7 (2.7)
Balance score, *n* (%)			
0	67 (2.0)	67 (2.0)	66 (2.0)
1	160 (4.8)	160 (4.8)	154 (4.6)
2	193 (5.7)	193 (5.7)	187 (5.6)
3	316 (9.4)	316 (9.4)	312 (9.3)
4	2812 (78.2)	2811 (78.2)	2786 (78.5)
Repeated chair stand score, *n* (%)			
0	385 (11.5)	385 (11.5)	376 (11.3)
1	372 (10.8)	372 (10.8)	364 (10.7)
2	535 (15.0)	535 (15.0)	526 (15.0)
3	798 (22.4)	797 (22.4)	794 (22.5)
4	1458 (40.2)	1458 (40.2)	1445 (40.4)
Gait score, *n* (%)			
0	124 (3.6)	124 (3.6)	121 (3.6)
1	129 (4.1)	129 (4.1)	124 (3.9)
2	273 (8.1)	273 (8.1)	268 (8.1)
3	554 (16.1)	553 (16.1)	544 (16.1)
4	2468 (68.1)	2468 (68.1)	2448 (68.4)
Mobility impairments (range: 0–10), mean (SD)	2.0 (2.4)	–	–
ADL disabilities (range: 0–6), mean (SD)	–	0.3 (0.8)	–
IADL disabilities (range: 0–7), mean (SD)	–	0.3 (0.8)	–
Falls, *n* (%)			
No	–	–	2439 (69.5)
Yes	–	–	1066 (30.5)

SD, standard deviation; *n*, number of participants; SPPB, Short Physical Performance Battery; BMI, body mass index; ADL, activities of daily living; IADL, instrumental activities of daily living. ^1^ Age was collapsed to 90 for participants aged 90+ years. Note: Unweighted frequencies and weighted percentages are presented. All other values are weighted estimates. While the same participants were included in the complete-case samples for models with ADL disabilities or IADL disabilities as outcomes (and the participant characteristics are therefore aggregated for this sample in Table 1), these were treated as distinct outcomes in the analyses.

**Table 2 ijerph-19-16319-t002:** Mean changes in mobility impairment, ADL disabilities, IADL disabilities, and the odds ratios of falls per one-point increase in the total SPPB score, using repeated measures outcomes from seven waves of follow-up.

	**Estimate (95% CI)**	***p*-Value**
Mobility impairment (*n* = 3548; 16,934 observations)		
Model 1: Univariate association	−0.55 (−0.58, −0.52)	<0.001
Model 2: Model 1 plus continuous linear time variable	−0.55 (−0.58, −0.52)	<0.001
Model 3: Model 2 plus continuous quadratic time variable ^1^	−0.55 (−0.57, −0.52)	<0.001
Model 4: Model 2 plus socio-demographic covariates ^2^	−0.50 (−0.53, −0.47)	<0.001
Model 5: Model 4 plus health-related covariates ^3^	−0.36 (−0.40, −0.33)	<0.001
Model 6: Model 5 plus mobility impairments at Wave 2	−0.13 (−0.16, −0.10)	<0.001
ADL disabilities (*n* = 3547; 16,934 observations)		
Model 1: Univariate association	−0.16 (−0.18, −0.15)	<0.001
Model 2: Model 1 plus continuous linear time variable	−0.16 (−0.18, −0.15)	<0.001
Model 3: Model 2 plus continuous quadratic time variable	−0.16 (−0.18, −0.15)	<0.001
Model 4: Model 3 plus socio-demographic covariates ^2^	−0.16 (−0.18, −0.14)	<0.001
Model 5: Model 4 plus health-related covariates ^3^	−0.12 (−0.14, −0.11)	<0.001
Model 6: Model 5 plus ADL disabilities at Wave 2	−0.06 (−0.08, −0.05)	<0.001
IADL disabilities (*n* = 3547; 16,934 observations)		
Model 1: Univariate association	−0.18 (−0.19, −0.16)	<0.001
Model 2: Model 1 plus continuous linear time variable	−0.17 (−0.19, −0.15)	<0.001
Model 3: Model 2 plus continuous quadratic time variable	−0.17 (−0.18, −0.15)	<0.001
Model 4: Model 3 plus socio-demographic covariates ^2^	−0.15 (−0.17, −0.13)	<0.001
Model 5: Model 4 plus health-related covariates ^3^	−0.11 (−0.13, −0.09)	<0.001
Model 6: Model 5 plus IADL disabilities at Wave 2	−0.06 (−0.07, −0.04)	<0.001
	**OR (95% CI)**	***p*-Value**
Falls (*n* = 3505; 16,332 observations)		
Model 1: Univariate association	0.85 (0.83, 0.87)	<0.001
Model 2: Model 1 plus continuous linear time variable	0.84 (0.82, 0.86)	<0.001
Model 3: Model 2 plus continuous quadratic time variable ^1^	0.84 (0.82, 0.86)	<0.001
Model 4: Model 2 plus socio-demographic covariates ^2^	0.87 (0.85, 0.89)	<0.001
Model 5: Model 4 plus health-related covariates ^3^	0.90 (0.88, 0.93)	<0.001
Model 6: Model 5 plus falls at Wave 2	0.92 (0.90, 0.95)	<0.001

ADL, activities of daily living; IADL, instrumental activities of daily living; SPPB, Short Physical Performance Battery; *n*, number of participants; CI, confidence interval; OR, odds ratio. ^1^ The quadratic time variable was subsequently excluded from the models, as there was no evidence of non-linearity. ^2^ Age, biological sex, ethnicity, marital status, employment status, education, and wealth. ^3^ Physical activity, body mass index, cognitive function, and depressive symptoms. Note: All values are weighted estimates.

**Table 3 ijerph-19-16319-t003:** Multilevel model results for SPPB domain scores at baseline (balance, repeated chair stands, and gait) on mobility impairment, ADL disabilities, IADL disabilities, and falls over seven waves of follow-up; each exposure was entered in separate models.

	Mobility Impairment ^1^		ADL Disabilities ^2^		IADL Disabilities ^2^		Falls ^3^	
	Estimate (95% CI)	*p*-Value	Estimate (95% CI)	*p*-Value	Estimate (95% CI)	*p*-Value	OR (95% CI)	*p*-Value
**Unadjusted**								
Balance								
0 (reference)	0.00		0.00		0.00		1.00	
1	−0.71 (−1.53, 0.10)	0.087	−0.84 (−1.32, −0.35)	0.001	−0.33 (−0.80, 0.13)	0.159	0.96 (0.56, 1.65)	0.877
2	−1.09 (−1.88, −0.30)	0.007	−1.15 (−1.62, −0.68)	<0.001	−0.77 (−1.19, −0.34)	<0.001	0.92 (0.55, 1.54)	0.756
3	−2.05 (−2.80, −1.30)	<0.001	−1.32 (−1.78, −0.86)	<0.001	−0.97 (−1.37, −0.56)	<0.001	0.60 (0.37, 0.98)	0.042
4	−3.53 (−4.22, −2.83)	<0.001	−1.69 (−2.13, −1.24)	<0.001	−1.43 (−1.80, −1.06)	<0.001	0.33 (0.21, 0.52)	<0.001
Repeated chair stands								
0 (reference)	0.00		0.00		0.00		1.00	
1	−1.99 (−2.36, −1.62)	<0.001	−0.77 (−0.94, −0.60)	<0.001	−0.69 (−0.90, −0.49)	<0.001	0.47 (0.36, 0.61)	<0.001
2	−2.83 (−3.16, −2.50)	<0.001	−0.98 (−1.14, −0.82)	<0.001	−0.88 (−1.06, −0.70)	<0.001	0.42 (0.33, 0.54)	<0.001
3	−3.38 (−3.69, −3.08)	<0.001	−1.12 (−1.27, −0.97)	<0.001	−1.03 (−1.20, −0.86)	<0.001	0.41 (0.32, 0.51)	<0.001
4	−4.17 (−4.45, −3.89)	<0.001	−1.26 (−1.40, −1.12)	<0.001	−1.20 (−1.36, −1.04)	<0.001	0.32 (0.26, 0.39)	<0.001
Gait								
0 (reference)	0.00		0.00		0.00		1.00	
1	0.25 (−0.46, 0.95)	0.489	−0.25 (−0.62, 0.11)	0.174	0.41 (−0.04, 0.85)	0.073	0.78 (0.49, 1.24)	0.294
2	−0.72 (−1.36, −0.08)	0.028	−0.56 (−0.87, −0.25)	<0.001	−0.22 (−0.60, 0.16)	0.255	0.64 (0.42, 0.96)	0.030
3	−2.41 (−3.01, −1.81)	<0.001	−1.05 (−1.33, −0.76)	<0.001	−0.77 (−1.13, −0.41)	<0.001	0.40 (0.28, 0.58)	<0.001
4	−3.94 (−4.51, −3.38)	<0.001	−1.39 (−1.67, −1.12)	<0.001	−1.23 (−1.58, −0.89)	<0.001	0.26 (0.18, 0.36)	<0.001
**Fully adjusted ^4^**								
Balance								
0 (reference)	0.00		0.00		0.00		1.00	
1	0.08 (−0.52, 0.67)	0.800	−0.32 (−0.67, 0.03)	0.074	0.00 (−0.35, 0.36)	0.992	0.91 (0.54, 1.54)	0.734
2	0.16 (−0.39, 0.72)	0.561	−0.52 (−0.86, −0.18)	0.002	−0.27 (−0.59, 0.05)	0.102	0.93 (0.57, 1.50)	0.757
3	0.04 (−0.51, 0.59)	0.878	−0.48 (−0.81, −0.16)	0.004	−0.25 (−0.56, 0.06)	0.118	0.76 (0.48, 1.21)	0.245
4	−0.33 (−0.85, 0.18)	0.205	−0.59 (−0.91, −0.27)	<0.001	−0.36 (−0.65, −0.06)	0.017	0.58 (0.37, 0.89)	0.013
Repeated chair stands								
0 (reference)	0.00		0.00		0.00		1.00	
1	−0.43 (−0.69, −0.16)	0.002	−0.29 (−0.43, −0.16)	<0.001	−0.25 (−0.41, −0.09)	0.002	0.57 (0.44, 0.73)	<0.001
2	−0.59 (−0.84, −0.34)	<0.001	−0.35 (−0.48, −0.22)	<0.001	−0.26 (−0.41, −0.12)	<0.001	0.66 (0.52, 0.84)	0.001
3	−0.61 (−0.86, −0.36)	<0.001	−0.38 (−0.50, −0.26)	<0.001	−0.26 (−0.41, −0.12)	<0.001	0.66 (0.53, 0.83)	<0.001
4	−0.83 (−1.08, −0.59)	<0.001	−0.41 (−0.53, −0.29)	<0.001	−0.31 (−0.44, −0.18)	<0.001	0.63 (0.51, 0.79)	<0.001
Gait								
0 (reference)	0.00		0.00		0.00		1.00	
1	−0.30 (−0.77, 0.17)	0.206	−0.31 (−0.57, −0.04)	0.024	−0.01 (−0.34, 0.32)	0.957	0.64 (0.42, 0.99)	0.044
2	−0.20 (−0.60, 0.21)	0.346	−0.27 (−0.50, −0.03)	0.025	−0.06 (−0.34, 0.22)	0.694	0.66 (0.45, 0.95)	0.026
3	−0.43 (−0.82, −0.05)	0.027	−0.37 (−0.59, −0.16)	0.001	−0.25 (−0.52, 0.01)	0.059	0.56 (0.40, 0.78)	0.001
4	−0.86 (−1.24, −0.49)	<0.001	−0.49 (−0.69, −0.29)	<0.001	−0.42 (−0.68, −0.17)	0.001	0.45 (0.32, 0.61)	<0.001

ADL, activities of daily living; IADL, instrumental activities of daily living; SPPB, Short Physical Performance Battery; CI, confidence interval; OR, odds ratio. ^1^ Number of participants: 3548. Total observations: 16,934. ^2^ Number of participants: 3547. Total observations: 16,934. ^3^ Number of participants: 3505. Total observations: 16,332. ^4^ Adjusted for linear time, quadratic time (excluded from the models with falls as the outcome, as there was no evidence of non-linearity), age, biological sex, ethnicity, marital status, employment status, education, wealth, physical activity, body mass index, cognitive function, depressive symptoms, and the outcome from Wave 2. Note: All values are weighted estimates.

## Data Availability

Restrictions apply to the availability of these data. Data were obtained from the UK Data Service (SN 5050) and are available at https://beta.ukdataservice.ac.uk/datacatalogue/studies/study?id=5050 (accessed on 1 June 2022). The Stata syntax used to replicate the analyses presented in this article is openly available from GitHub at https://github.com/OliviaMalkowski/SPPB-validity.git (accessed on 1 June 2022).

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
