# Peer review of "Associations of the Short Physical Performance Battery (SPPB) with Adverse Health Outcomes in Older Adults: A 14-Year Follow-Up from the English Longitudinal Study of Ageing (ELSA)"

_ijerph, 2022, doi:10.3390/ijerph192316319_

Round 1

Reviewer 1 Report

The paper is well written und interesting. I would only suggest to split Table 1 into 2 separate tables (one with the SPPB results and one with the other results).

Congratulations to this fine paper

Author Response

Thank you for the positive feedback on our paper. We considered your suggestion to split Table 1 into two separate tables, however felt this might not be necessary given that it would result in two consecutive tables presenting baseline data for the sample with no/little intermediary text.

Reviewer 2 Report

Thank you for the opportunity to review this piece of work.

Major comment:

I noticed that the aim of the study is written as: The aim of the present study was therefore to examine the predictive value of the SPPB and its domains of balance, strength, and gait speed on key functional outcomes collected over 14 years. I saw the statistical analysis and results section, saw the numbers of models generated but don't see the R2 or goodness of fit which indicates predictive modelling. It looks more like finding out association between SPPB and outcomes which is reflected by logistic and linear regressions. My concern is whether the authors are looking for association between SPPB and outcomes or using SPPB as a predictor for the outcomes which are totally different research questions.

Minor comment: noticed a typo at line 115, I assume it's divided by... and not divided my...

Author Response

Reviewer 2:

Thank you for the opportunity to review this piece of work.

Major comment:

I noticed that the aim of the study is written as: The aim of the present study was therefore to examine the predictive value of the SPPB and its domains of balance, strength, and gait speed on key functional outcomes collected over 14 years. I saw the statistical analysis and results section, saw the numbers of models generated but don't see the R2 or goodness of fit which indicates predictive modelling. It looks more like finding out association between SPPB and outcomes which is reflected by logistic and linear regressions. My concern is whether the authors are looking for association between SPPB and outcomes or using SPPB as a predictor for the outcomes which are totally different research questions.

Thank you for your feedback and helpful suggestions.

Thank you for this comment. Given the challenges of generating marginal and/or conditional R2 values for mixed-effects models in Stata, as well as inconsistencies in the way that R2 is defined for mixed-effects models, these values were not generated. We agree that this is an association rather than a predictive study and that any terminology indicative of prediction could be misleading to a reader. Any mentions of “predictive validity” or predictions have been replaced with terms suggestive of an association. This includes changes to the title (lines 2-4), abstract (lines 11 and 21), and main text (lines 62-64, 67-68, 221, 585, 592-593, 782, 789-790, 802, 818-819, 826, 862-869, and 940).

Minor comment: noticed a typo at line 115, I assume it's divided by... and not divided my...

Thank you for noticing this and bringing it to our attention. This has now been corrected.

Reviewer 3 Report

This paper describes the associations between SPPB and its separate components and various health outcomes based on data from the ELSA study. Overall this is a very clearly written paper with extensive and sound statistical analyses. I have a few comments:

Only few papers regarding the validity of the SPPB measure have been added to the introduction and discussion paragraphs while more exist. I suggest to add a few more with outcomes that correspond with the outcomes of the current paper. 

In 2.2.1. in the first sentence, it reads "four outcomes", but I only discovered three outcomes. Perhaps ADL and IADL are listed here as separate outcomes? This needs some clarification. 

The SPPB measurements and scoring are very well explained in the supplementary figures. I was wondering if these measurements correspond to the original SPPB measurements by Prof. Guralnik or whether some modifications were made. I guess this is the case for the balance test which is different for persons below the age of 70 and above (10 vs. 30 seconds). It would be interesting to discuss these adjustments in the discussion paragraph: how does this affect your results?

The statistical analyses are done very thoroughly, but still easy to follow. 

The numbers in Table 1 do not always add up to the totals, for example the numbers regarding education level are similar for mobility impairments and ADL/IADL outcome groups, while the ADL/IADL outcome should have 1 respondent less compared to the mobility impairments outcome. Perhaps more errors are present. 

Zero-inflated negative binomial models are recommended for these types of analyses but were not considered due to software constraints. I am not familiar with this type of analyses. What are the potential implications of this limitation?

Author Response

Reviewer 3:

This paper describes the associations between SPPB and its separate components and various health outcomes based on data from the ELSA study. Overall this is a very clearly written paper with extensive and sound statistical analyses. I have a few comments:

Only few papers regarding the validity of the SPPB measure have been added to the introduction and discussion paragraphs while more exist. I suggest to add a few more with outcomes that correspond with the outcomes of the current paper.

Thank you for your positive feedback and the useful suggestions.

Please find an additional paragraph in the introduction section (lines 59-66) to address this and an expanded discussion in lines 862-872. The reference list and corresponding numeric in-text citations have been updated accordingly.

In 2.2.1. in the first sentence, it reads "four outcomes", but I only discovered three outcomes. Perhaps ADL and IADL are listed here as separate outcomes? This needs some clarification.

Thank you for the suggestion. We have now updated section 2.2.1. by numbering the four outcomes for clarity in lines 89-91. An additional note has also been added beneath Table 1 to explain that, while the complete-case samples for the models with ADL disabilities and IADL disabilities as outcomes contained data from the same sample of participants, these represent distinct outcomes in the statistical analyses.

The SPPB measurements and scoring are very well explained in the supplementary figures. I was wondering if these measurements correspond to the original SPPB measurements by Prof. Guralnik or whether some modifications were made. I guess this is the case for the balance test which is different for persons below the age of 70 and above (10 vs. 30 seconds). It would be interesting to discuss these adjustments in the discussion paragraph: how does this affect your results?

Thank you for the comment on the supplementary figures and the reflection regarding modified versions of the SPPB. Although there were a few minor “subjective” decisions that we made as authors to accommodate the way in which the SPPB data was recorded in the ELSA dataset (such as when to assign a participant a missing value versus a zero – these decisions are outlined in the supplementary figures), the majority of the scoring is directly aligned to Professor Guralnik’s original work. The measured tests themselves in ELSA also directly conform to the original guidelines set out in Professor Guralnik’s paper, with the exception of the full-tandem stand, where respondents aged 69 years and under were asked to attempt the stand for 30 seconds. For the purpose of the present study, we collapsed data from this “modified” balance test, such that the scoring and results from the reported test would still match the original measures. As such, we are not able to deduce from our analyses whether/how using a different scoring system to account for the fact that some participants held for more than 10 seconds would influence our results.

The statistical analyses are done very thoroughly, but still easy to follow.

The numbers in Table 1 do not always add up to the totals, for example the numbers regarding education level are similar for mobility impairments and ADL/IADL outcome groups, while the ADL/IADL outcome should have 1 respondent less compared to the mobility impairments outcome. Perhaps more errors are present.

Thank you for the positive feedback on the statistical analyses.

The frequencies in Table 1 originally reflected the weighted frequencies, which contained decimal places. These numbers were rounded to the nearest integer, and therefore the sum of the frequencies for each variable did not always amount to the total “exact” number of participants in each sample. Upon reflection, we believe it is more useful to present unweighted raw frequencies and weighted proportions to avoid confusion; these numbers have been updated in Table 1. The note beneath Table 1 has also been amended to reflect these changes.

Zero-inflated negative binomial models are recommended for these types of analyses but were not considered due to software constraints. I am not familiar with this type of analyses. What are the potential implications of this limitation?

Thank you for this comment. To address your question, we have added an additional sentence to expand on the potential implications of this limitation in lines 935-938.

Round 2

Reviewer 2 Report

No further questions from me. Thanks.